# Elucidating the active phases of CoO$_x$ films on Au(111) in the CO oxidation reaction

Hao Chen [1], Lorenz J. Falling[2,3], Heath Kersell[2,4], George Yan[5], Xiao Zhao [3,6], Judit Oliver-Meseguer [1], Max Jaugstetter[1], Slavomir Nemsak [2,7], Adrian Hunt[8], Iradwikanari Waluyo [8], Hirohito Ogasawara [9], Alexis T. Bell [1,10], Philippe Sautet [5,11] & Miquel Salmeron [3,6] ✉

Noble metals supported on reducible oxides, like CoO$_x$ and TiO$_x$, exhibit superior activity in many chemical reactions, but the origin of the increased activity is not well understood. To answer this question we studied thin films of CoO$_x$ supported on an Au(111) single crystal surface as a model for the CO oxidation reaction. We show that three reaction regimes exist in response to chemical and topographic restructuring of the CoO$_x$ catalyst as a function of reactant gas phase CO/O$_2$ stoichiometry and temperature. Under oxygen-lean conditions and moderate temperatures (≤150 °C), partially oxidized films (CoO$_{x<1}$) containing Co$^0$ were found to be efficient catalysts. In contrast, stoichiometric CoO films containing only Co$^{2+}$ form carbonates in the presence of CO that poison the reaction below 300 °C. Under oxygen-rich conditions a more oxidized catalyst phase (CoO$_{x>1}$) forms containing Co$^{3+}$ species that are effective in a wide temperature range. Resonant photoemission spectroscopy (ResPES) revealed the unique role of Co$^{3+}$ sites in catalyzing the CO oxidation. Density function theory (DFT) calculations provided deeper insights into the pathway and free energy barriers for the reactions on these oxide phases. These findings in this work highlight the versatility of catalysts and their evolution to form different active phases, both topological and chemically, in response to reaction conditions exposing a new paradigm in the catalyst structure during operation.

Catalysts are defined as materials that facilitate chemical reactions by providing special sites where reactants and products bind, react, and desorb with low energy barriers separating these steps. Although the composition and structure of the catalyst is usually assumed to be unaltered during reaction, many catalysts restructure by displacement of its atoms in response to the adsorption of reactants[1–3]. Here we demonstrate that in addition to topographic restructuring, chemical restructuring can also occur, adding an additional paradigm in the understanding of the working catalysts. In this work we illustrate this by showing that CoO$_x$ catalysts for the CO oxidation reaction undergo both chemical and topological changes in response to the reactant gas CO/O$_2$ stoichiometry, evolving in three regimes characterized by

[1]Chemical Sciences Division, Lawrence Berkeley National Laboratory, Berkeley, CA 94720, USA. [2]Advanced Light Source, Lawrence Berkeley National Laboratory, Berkeley, CA 94720, USA. [3]Materials Sciences Division, Lawrence Berkeley National Laboratory, Berkeley, CA 94720, USA. [4]School of Chemical, Biological, and Environmental Engineering, Oregon State University, Corvallis, OR 97331, USA. [5]Department of Chemical and Biomolecular Engineering, University of California, Los Angeles, Los Angeles, CA 90095, USA. [6]Department of Materials Science and Engineering, University of California, Berkeley, CA 94720, USA. [7]Department of Physics and Astronomy, University of California, Davis, CA 95616, USA. [8]National Synchrotron Light Source II, Brookhaven National Laboratory, Upton, NY 11973, USA. [9]SLAC National Accelerator Laboratory, 2575 Sand Hill Road, Menlo Park, CA 94025, USA. [10]Department of Chemical and Biomolecular Engineering, University of California, Berkeley, CA 94720, USA. [11]Department of Chemistry and Biochemistry, University of California, Los Angeles, Los Angeles, CA 90095, USA. ✉e-mail: mbsalmeron@lbl.gov

different Co oxidation states. Partially oxidized films of cobalt ($CoO_{x<1}$) deposited on Au containing $Co^0$ were found to efficiently catalyze the CO oxidation reaction under oxygen-lean conditions and at temperatures lower than 150 °C. With increasing $O_2$ content CoO forms first, which reacts with CO to form carbonates that poison the reaction for temperatures below 300 °C. Finally, under oxygen-rich conditions, more oxidized phases ($CoO_{x>1}$) containing $Co^{3+}$ species form that are more effective catalysts in a broad temperature range. Ambient Pressure X-ray photoelectron spectroscopy (APXPS) was employed to follow the catalyst oxidation state, the adsorption of CO and reaction products. While the various Co oxidation states can be monitored by APXPS during reaction, a precise identification and quantification is challenging due to the strong overlap of their 2p core level peaks. We overcome this difficulty using Resonant Photoelectron Spectroscopy (ResPES), which allowed us to precisely identify each Co oxidation state and relative concentration. DFT calculations provided in-depth perspectives related with CO oxidation reaction pathway and energy barriers on each $CoO_x$ phase.

## Results and discussion

### Deposition, oxidation, and wetting of cobalt films on Au(111)

Co2p XP spectra from a 1 MLE Co film on Au(111) before and after oxidation are shown in Fig. 1a. The bottom spectrum (black trace) displays the result before oxidation, showing the Co 2p3/2 level peak at 778.2 eV characteristic of metallic Co. After exposing the film to $10^{-6}$ Torr of $O_2$ gas at RT for 60 s (=60 Langmuir units), a partially oxidized Co film was formed. This is shown in the red spectrum by the additional peak at 780.2 eV, strongly overlapping with the $Co^0$ peak, and its satellite at 786.6 eV, both characteristic of $Co^{2+}$. Fitting the two overlapping $2p_{3/2}$ peaks (Fig. S1), we estimate that ~25% of the Co atoms are oxidized to $Co^{2+}$. We will refer to this film as $CoO_{0.25}$. Annealing the film in $1 \times 10^{-6}$ Torr $O_2$ at 200 °C for 10 min led to the formation of CoO, with the peak at 780.2 eV from $Co^{2+}$ now being dominant (Fig. 1a, blue trace). Figure 1b shows the corresponding O 1s XPS region, with the lattice oxygen peak at 529.6 eV, for both $CoO_{0.25}$ and CoO. The peak at 531.2 eV is due to adsorbed OH and CO from residual background $H_2O$

and CO gases[4]. Fig. 1c shows XPS of the Au 4 f and Co3p region after each of these treatments. With an incident photon energy of 260 eV, the kinetic energy of photoelectrons exiting from the Au surface is ~180 eV, with an inelastic mean free path of ~ 5 Å[5] i.e., ~ 2 atomic layers. After deposition of 1 MLE of metallic Co (black trace in Fig. 1c), the Au 4 f peak intensity decreased by ~ 40% compared to the pristine Au(111) (gray trace in Fig. 1c). This attenuation is consistent with the double-layer island structure of metallic cobalt (see SI) as described by Morgenstern et al.[6]. After exposing this film to $1 \times 10^{-6}$ Torr $O_2$ and annealing to 200 °C (blue trace in Fig. 1c), the Au peak intensity decreased to ~ 20% of its clean surface value. This attenuation is consistent with the spreading of $CoO_x$ and can be described well by a layer-by-layer growth (Figs. S2 and S3).

Further oxidation at 200 °C in a 105 mTorr of an $O_2$-rich reaction mixture ($O_2/CO = 20:1$) completed the oxidation of cobalt to $Co^{2+}$ and $Co^{3+}$, characterized by peaks at 780.2 and 779.8 eV respectively (green trace in Fig. 1a). The $Co^{3+}$ is further characterized by the increased intensity of the satellite at 789 eV. We will refer to this film as $Co_3O_4$[7]. The presence of the 2+ and 3+ oxidation states of Co and their proportion in the film will be confirmed and quantified later using ResPES.

### CO adsorption on $CoO_{x<1}$ and on $CoO_{x=1}$

Figure 2a shows the C 1s XPS region from the $CoO_{0.25}$ film before and after introduction of 100 mTorr of CO. Before CO introduction (bottom gray trace), the sample shows a C 1s peak at 284 eV, due to adventitious C contamination[4]. In the presence of 100 mTorr of CO, a strong peak at 286 eV appears due to chemisorbed molecular CO (red trace), which adsorbs only on metallic $Co^0$ sites[8,9]. The small peak at 289 eV is due to carbonate species formed on the CoO areas occupying ~25% of the oxide film area. A peak from gas-phase CO, with its fine vibrational structure is visible at 291.6 eV. After pumping out the CO gas, the $CO_{ads}$ peak nearly vanished (top gray trace), due to desorption to equilibrate with the reduced gas pressure. By contrast, on a CoO film under 100 mTorr CO at RT only the peak at 289 eV associated with carbonate species is observed (Fig. 2b, middle curve), along with the peak of gas-phase CO at 292 eV, but no molecularly adsorbed CO is

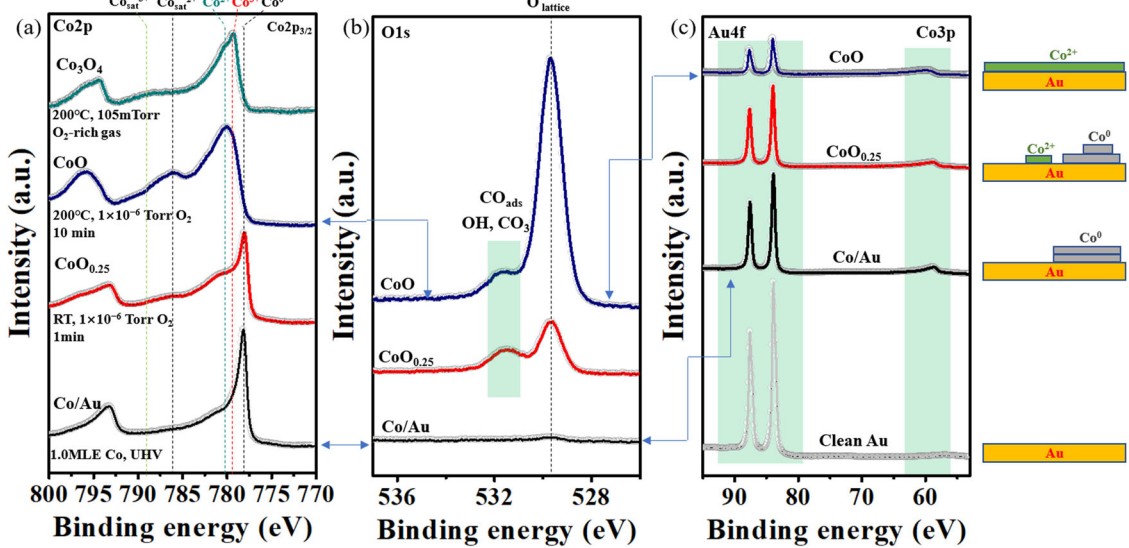

**Fig. 1 | Oxidation and wetting of cobalt films deposited on Au(111). a** From the bottom: Co2p spectra of 1 MLE $Co^0$ (black curve); after room temperature exposure to $1 \times 10^{-6}$ Torr $O_2$ for 1 min ($CoO_{0.25}$, red); after oxidation under $1 \times 10^{-6}$ Torr $O_2$ at 200 °C for 10 min (CoO, blue); and under 105 mTorr of $O_2$-rich gas conditions ($CO:O_2 = 1:20$) at 200 °C ($Co_3O_4$, green). The dotted vertical lines mark the binding energy positions of the $2p_{3/2}$ core levels of metallic Co ($Co^0$), CoO ($Co^{2+}$), $Co_3O_4$ ($Co^{2+}$ and $Co^{3+}$), and their shake-up peak satellites ($Co_{sat}^{2+}$ and $Co_{sat}^{3+}$). **b** Corresponding O 1s core level regions. From the bottom: mostly clean metallic

Co (black); $CoO_{0.25}$ film (red); and after 10 min exposure to $1 \times 10^{-6}$ Torr $O_2$ at 200 °C (blue). **c** Au4f core level spectral region. From bottom: clean Au (gray curve); after deposition of 1 MLE of Co (black curve); after annealing under $1 \times 10^{-6}$ Torr $O_2$ at 200 °C for 10 min (red); and after 10 min exposure to $1 \times 10^{-6}$ Torr $O_2$ at 200 °C (blue). The strong decrease of the Au 4 f peak intensity is due to the spreading of the $CoO_x$ film, now covering more Au surface than the initial metallic film. On the right is a schematic illustration of the wetting process during the oxidation process. The XPS data were acquired at beamline 23-ID-2 (IOS) of NSLS-II.

observed. The area of the carbonate peak on the CoO film became about 4 times larger than that on the $CoO_{0.25}$ film. The amount of carbonates is substantially increased by exposing the sample to $CO_2$ instead of CO, as shown in the top trace in Fig. 2b.

## CO oxidation reaction catalyzed by partially oxidized cobalt ($CoO_{x<1}$)

In the previous section we identified the species formed by CO adsorption on $CoO_{x<1}$ and $CoO_{x=1}$ at RT. Here we follow the evolution of the $CoO_{0.25}$ surface during the CO oxidation via the Mars van-Krevelen mechanism as a function of temperature. The surface composition, followed by APXPS, is shown in Fig. 3. In UHV and in 100 mTorr of CO at RT the spectra in the C 1s region are similar to those in Fig. 2. Heating to 100 °C caused a decrease in the intensity of the C 1s peak at 286 eV from adsorbed CO due to the new equilibrium with the gas phase at the higher temperature (Fig. 3b). This is shown by

the O lattice peak at 529.6 eV and the $Co^0$ peak at 778 eV (Fig. 3a, c), which remained essentially unchanged at this temperature. The O 1s peak at 531.8 eV, responding to overlapping O peaks from adsorbed CO and OH, decreased due to thermal desorption. The new small peak at 283.0 eV originates from cobalt carbide ($CoC_x$), suggesting CO dissociation at $Co^0$ sites at elevated temperature[8]. Raising the temperature to 150 °C increased the reaction rate, as shown by the increase of the $Co^0$ peak at 778.2 eV, the decrease of Co $2p_{3/2}$ peak at 780.2 eV from $Co^{2+}$ (Fig. 3a), and the decrease of the lattice oxygen peak at 529.6 eV. All these changes confirm the reduction of $CoO_{0.25}$ by reaction between CO and lattice oxygen.

## CO oxidation reaction on cobalt monoxide

After oxidation of 1 MLE of Co to form CoO, with all cobalt atoms in the $Co^{2+}$ state, CO was introduced in the chamber to a pressure of 100 mTorr. The reaction was monitored by APXPS as a function of temperature, with the results shown in Fig. 4. The spectra show that the intensity of the $Co^{2+}$ in Fig. 4a and the O peaks in Fig. 4c remained largely unchanged up to 300°C but dropped rapidly thereafter. The decrease of these peaks can be attributed to the combined oxide reduction upon decomposition of the carbonate, and cobalt monoxide (CoO) deweting, that creates CoO clusters and exposes more Au surface, as shown by the rapid increase of the Au 4 f peak intensity in Fig. 4d and illustrated schematically in the inset. The deweting is the result of the formation of unstable carbonates, and reveals a topographical restructuring accompanying the change in oxidation state phase, from $CoO_{x=1}$ to $CoO_{x<1}$. Details of the reaction are shown in SI (Figs. S4 and S5). Introduction of 100 mTorr $O_2$ at ~350 °C caused the reappearance of the Co2p peak (SI, Fig. S6a), indicating no loss of Co by formation of Co-carbonyls. We discuss this reaction mechanism further in the theoretical studies in below Section 3.6.

## CO oxidation catalyzed by $CoO_{x>1}$

In the previous sections, the oxide film contained $Co^{\delta+}$ species with $\delta \le 2$. However, under oxygen-rich reaction conditions, oxide phases including O-Co-O trilayers[7,10] or $Co_3O_4$ phase[11], containing $Co^{3+}$ are present, which has been proposed by several authors to be the most active site for CO oxidation[12-15]. To our knowledge, however, this has not been proven spectroscopically in operando conditions. To ascertain this important point, it is necessary to unambiguously distinguish spectroscopically the Co oxidation states, $Co^{3+}$ and $Co^{2+}$, both involving different partially filled and empty Co d-levels. This can be

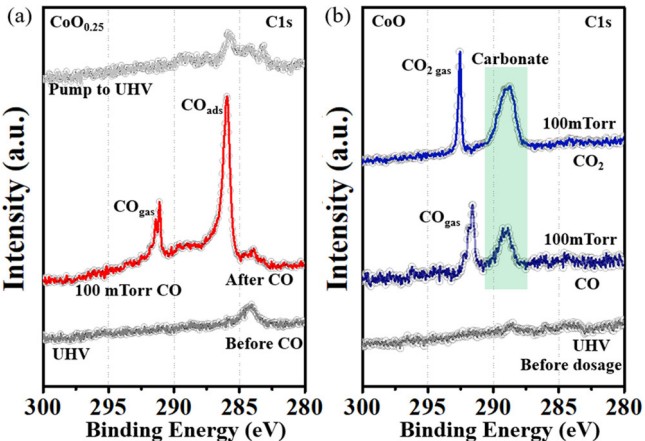

**Fig. 2 | Room Temperature CO adsorption on 1 MLE of $CoO_{0.25}$ (a) and CoO (b) on Au(111). a** C 1s XPS region of $CoO_{0.25}$ in UHV (gray, bottom); under 100 mTorr of CO (red, middle), and after pumping out the gas (gray, top); (**b**) C1s XPS region of CoO in UHV (gray, bottom), under 100 mTorr CO (blue, middle) and 100 mTorr $CO_2$ (blue, top). Molecular CO adsorbs only on metallic Co, and forms carbonates on the CoO regions. The amount of carbonate increases substantially in the presence of $CO_2$ gas (top trace). The XPS data were acquired at beamline 23-ID-2 (IOS) of NSLS-II.

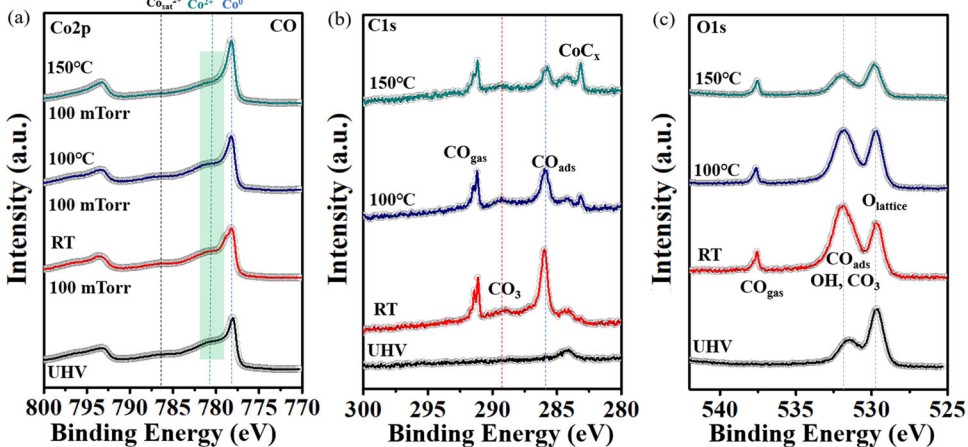

**Fig. 3 | Surface composition during CO oxidation on a partially oxidized cobalt film on Au(111).** APXPS in the (**a**) Co 2p, (**b**) C 1 s, and (**c**) O 1s core level regions during the CO oxidation reaction on Au-supported 1 MLE $CoO_{0.25}$ film. From bottom: at RT in UHV (black), under 100 mTorr of CO at RT (red), at 100 °C (blue), and at 150 °C (green). Above 100 °C the reduction of $Co^{2+}$ in the $CoO_{0.25}$ film is shown

by the increase in the Co $2p_{3/2}$ peak and the decrease of the lattice oxygen peak near 530 eV. The reaction rate is still low due to the presence of carbonate (peak at ~289 eV) that blocks the reaction. The XPS data were acquired at beamline 23-ID-2 (IOS) of NSLS-II.

done by Resonant Photoelectron Spectroscopy (ResPES), as proposed and demonstrated by several groups[16,17]. Briefly, ResPES is based on the photoemission of electrons from d-band states, enhanced by the resonant excitation of core-level electrons to empty d-states that decay by an Auger process of energy equal to the initial X-ray. This is

illustrated on the left panel of Fig. 5 for one of the oxidation states[18,19]. As the electronic configuration differs between $Co^{3+}$ ([Ar]$3d^6$) and $Co^{2+}$ ([Ar]$3d^7$), different resonant excitation energies for 2p to 3d transitions exist for each species. Since the Auger-mediated emission and the direct photoemission are undistinguishable processes with iden-

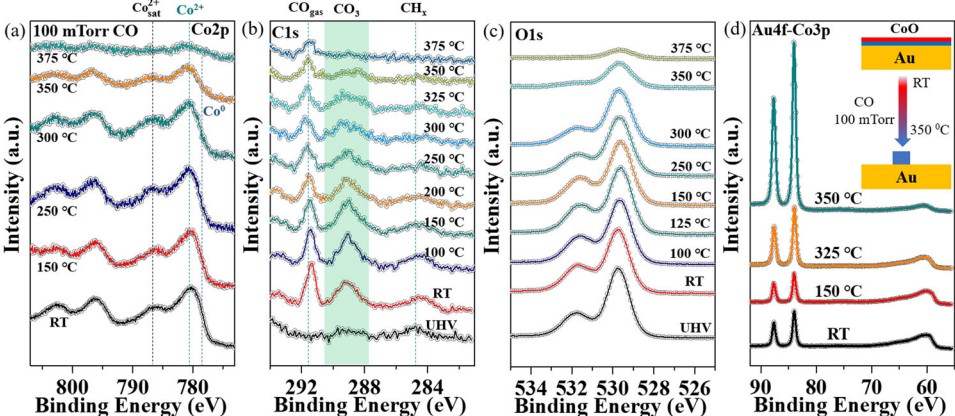

**Fig. 4 | Reduction and de-wetting of CoO by reaction with CO gas.** Reduction and de-wetting of CoO/Au by reaction with CO: (**a**–**c**) Co 2p, C 1 s, O 1 s, and Au 4f XP spectra of 1 MLE of CoO on Au. From bottom: in UHV (black), and under 100 mTorr CO after heating to the temperatures indicated. The decrease in the

intensity of the Co and O peaks above 300 °C is related to reduction, clustering, and dewetting of the unstable carbonate covered CoO, as shown by the rapid increase of the Au 4 f peak shown in **d**. Inset: graphic illustration of the dewetting process. The XPS data were acquired at beamline 9.3.2 of ALS.

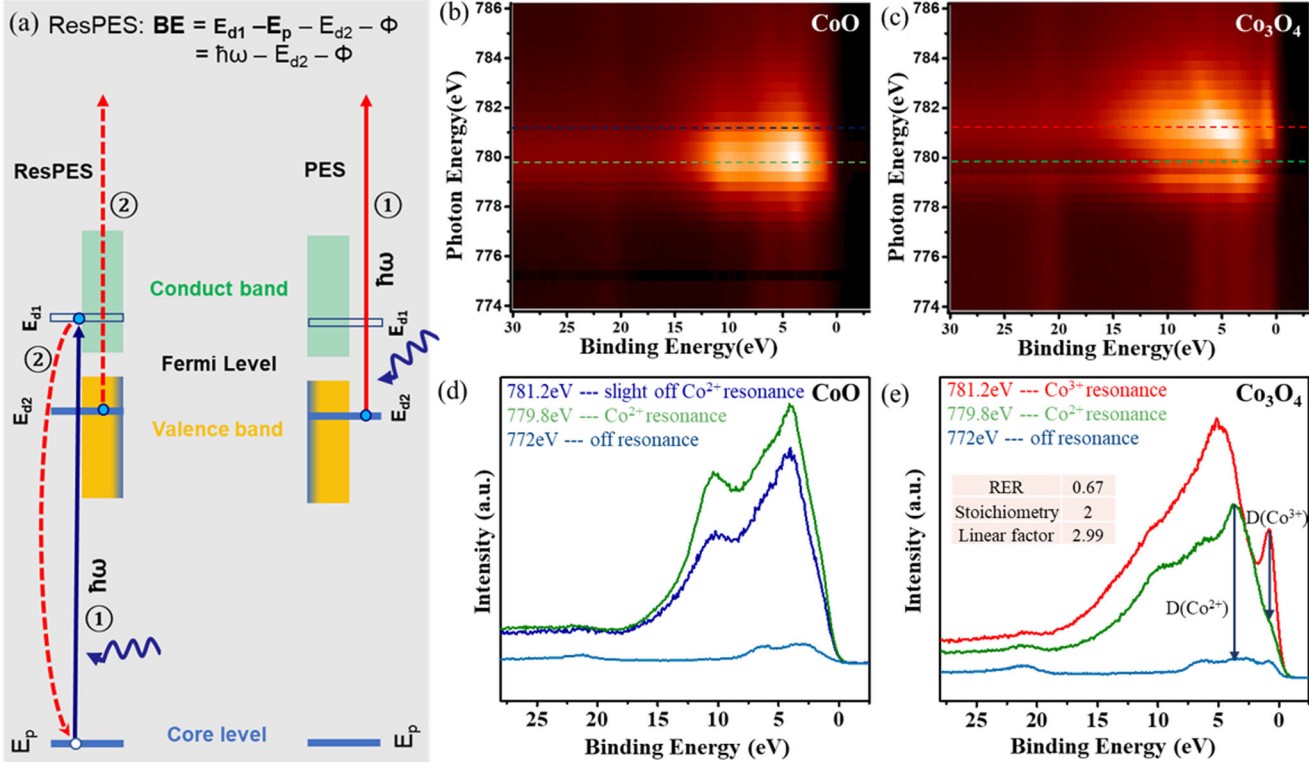

**Fig. 5 | Resonant Photoelectron Spectroscopy of CoO and Co₃O₄. a** Illustration of the ResPES process: a photon excites an electron from a core level $E_p$ to an unoccupied d1 level (blue arrow, left) belonging to a particular Co oxidation state, which decays by an Auger process (red lines) ejecting an electron from an occupied d2 state of that ion (left). The same final state can be obtained by direct absorption of the resonant photon by electrons in the same d2 level (right). These two processes are undistinguishable and therefore, interfere to enhance the transition. **b**, **c** Photoemission heat maps of Valence Band (VB) photoemission states from 1 MLE of CoO (**b**), and of 1 MLE of Co₃O₄ (**c**) on Au(111). (X = binding energy, Y =

exciting photon energy, Z = color-coded photoemission intensity). **d** VB spectra of CoO for ℏω = 779.8 eV, the resonant photon energy for the $Co^{2+}$ (green trace), for ℏω = 781.2 eV, slightly off resonance for the same state (blue trace); and for ℏω = 772 eV (bottom light blue trace), dominated by the d-levels of the Au substrate. **e** VB spectra of Co₃O₄ for ℏω = 781.2 eV, resonant photon energy for the $Co^{3+}$ state (red trace); for ℏω = 779.8 eV, resonant energy for the $Co^{2+}$ (green trace); and for ℏω = 772 eV off-resonance (bottom light blue trace), dominated by the d-levels of the Au substrate. Data were acquired at beamline 9.3.2 of ALS.

tical final state, a strong resonant enhancement of the photoemission spectra is observed. The photon energy for the resonant excitation can be experimentally determined by collecting valence band photoemission spectra as a function of X-ray energy. A maximum emission will be obtained at the resonant energy.

The experimental determination of the resonant energies for 1 MLE of $Co_3O_4$, containing $Co^{3+}$ and $Co^{2+}$ on Au is shown in the heat maps (bright to dark for high and low intensity) of Fig.5b, c. For CoO, only $Co^{2+}$ species are present, with a resonant photon energy of $\hbar\omega = $ 779.8 eV, as shown in the heat map. The VB d-states of $Co^{2+}$, with peaks at ~5 eV and ~10 eV, are strongly enhanced at this photon energy (green trace in (Fig. 5d). For $\hbar\omega = 781.2$ eV, slightly off resonance, the spectrum is similar as expected, but less intense (blue trace), and for $\hbar\omega = 772$ eV (far from resonance), the VB spectrum (light blue trace) is dominated by the Au substrate. For $Co_3O_4$ (Fig. 5c) the resonant photon energy for the $Co^{3+}$ site is 781.2 eV, as shown by the maximum in the heat map. The VB spectrum at this photon energy shows several $Co^{3+}$ d-band peaks: a sharp one at 1.0 eV, and others around 10 eV, and 5.0 eV (Fig. 5e). The VB of the Au substrate at approx. 1, 2.5, and 6 eV[20].

The contribution from the $Co^{3+}$ and $Co^{2+}$ ions, $D(Co^{3+})$ and $D(Co^{2+})$ in Fig. 5e, can be quantified by the difference in peak intensities relative to the $Co^{2+}$ and to the off-resonance spectra respectively (length of the arrow lines in Fig. 5e), and can be used to determine the relative concentration of these species because the resonant enhancement ratio (RER), $D(Co^{3+})/D(Co^{2+})$, is directly proportional to the ratio of the concentrations[16], $N(Co^{3+})/N(Co^{2+})$, with a linear correction factor (y) that can be determined from the known ratio in stoichiometric $Co_3O_4$ films, through the equation:

$$\frac{N(Co^{3+})}{N(Co^{2+})} = y*\frac{D(Co^{3+})}{D(Co^{2+})} = y*RER$$

The RER of stoichiometric $Co_3O_4$ is 0.67. Note that the contribution of the d levels from the Au substrate is subtracted to get the $D(Co^{2+})$, while the contribution of both $Co^{2+}$ and Au are subtracted at ~1 eV BE to get $D(Co^{3+})$, as respectively indicated by two vertical arrows

in Fig. 5e. Since the $N(Co^{3+})/N(Co^{2+})$ value for stoichimetric $Co_3O_4$ ($Co_2O_3$-CoO) is 2, the linear factor y = 2.99. Therefore, we can determine the concentration ratio, $N(Co^{3+})/N(Co^{2+})$ of nonstoichiometric $CoO_x$ film through the measurement of the RER if it contains both $Co^{3+}$ and $Co^{2+}$ sites.

The ResPES spectra in Fig. 6a–d demonstrate the role of $Co^{3+}$ species in 0.5 MLE of $CoO_{x>1}$ on Au(111) in the CO catalytic oxidation reaction. The ResPES in Fig. 6a corresponds to the initial film in UHV where the film is not completely oxidized (Fig. 6e, black curve). Exposure to 100 mTorr $O_2$ at RT, increased the peak of $Co^{3+}$ at 778.6 eV in the XPS (Fig. 6e, red curve), more clearly evidenced by the rise of the characteristic resonant peaks at 10 eV, 5.0 eV and 1.0 eV in Fig. 6b. The RER of this intermediate $CoO_x$ is 0.3 and, therefore its $N(Co^{3+})/N(Co^{2+})$ ratio is nearly unity. The oxide intermediates remained structurally stable at RT when the reaction conditions were switched to 100 mTorr CO (Fig. S6b, c). The nonstoichiometric cobalt oxide was reduced to CoO when the reaction temperature was raised to -100 °C, which is clearly seen in the ResPES of Fig. 6c. The reduction of $CoO_x$ at 100 °C indicates the high reactivity of the intermediate oxide phase. Importantly, adding 100 mTorr of $O_2$ to the 100 mTorr of CO at 100°C regenerated the intermediate phase as indicated by the reappearance of the $Co^{3+}$ resonant peak at ~1.0 eV (Fig. 6d) and the red shift of the Co $2p_{3/2}$ peak (Fig. 6e, green curve). Since this nonstoichiometric oxide phase is structurally stable under CO and $O_2$ mixtures ($CO/O_2 = 1$), it appears to be the most catalytically active phase for CO oxidation at mild temperatures in oxygen-rich conditions. From its RER value of 0.47, the $N(Co^{3+})/N(Co^{2+})$ ratio of this active phase is 1.4, indicating a higher ratio of $Co^{3+}$ species at 100 °C compared with that at RT. Furthermore, the stoichiometry parameter "x" in this intermediate $CoO_x$ is 1.29, namely $CoO_{1.29} \approx Co_3O_{3.87}$, which is structurally equal to the $Co_3O_4$ phase. $Co^{3+}$ species were also proposed to be the active sites in $Co_3O_4$ nanorods for the cryogenic CO oxidation reaction[14]. Recent surface science results have also confirmed the facile activation of molecular $O_2$ to peroxide ($O_2^{2-}$) and superoxide species ($O_2^-$) at the oxygen vacancies on the surface of $Co_3O_4(100)$ model catalysts[15]. This suggests that the newly-formed $Co^{3+}$ in the reactive environment,

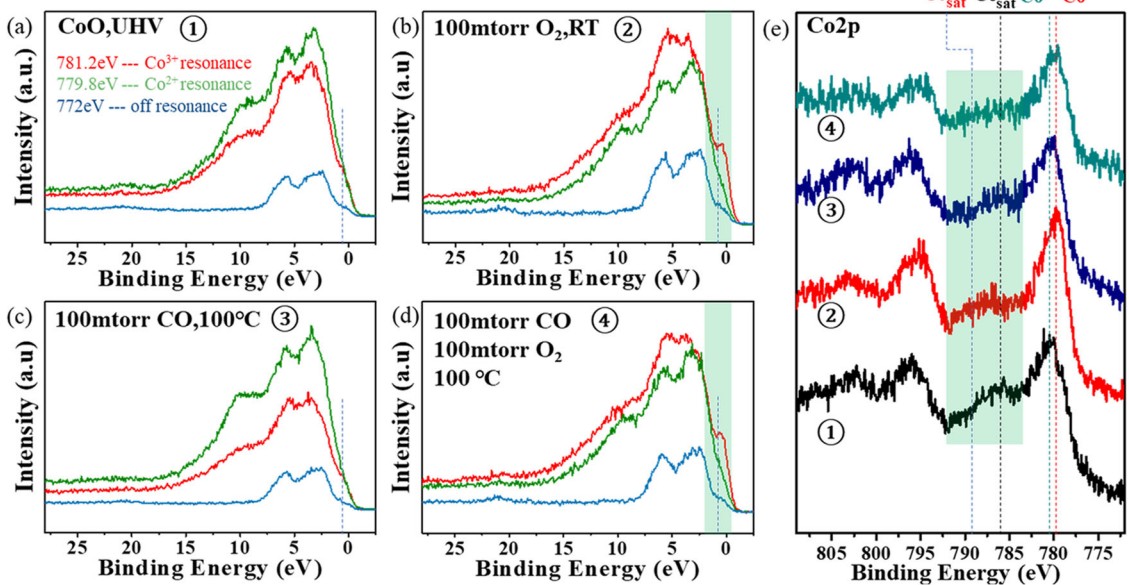

**Fig. 6 | Resonance Photoemission Spectra (ResPES) collected during the CO oxidation reaction on a 0.5 MLE $CoO_x$ on Au(111) sample. a–d** ResPES for an X-ray of energy $\hbar\omega = 781.2$ eV ($Co^{3+}$ resonance, red curve), for $\hbar\omega = 779.8$ eV ($Co^{2+}$ resonance, green curve) and for 772 eV (off-resonance, blue curve). **e** Co 2p XP spectra of CoO in UHV (black curve); under 100 mTorr $O_2$ at RT (red curve); under 100 mTorr CO at 100°C (blue curve); and under 100 mTorr CO and 100 mTorr $O_2$

(green curve). In (**a**) the dashed line marks the $Co^{3+}$ resonant peak position in the valence band. The intensity of the resonant peak increases from **a** to **b** due to oxidation, and decreases (**c**) due to reduction by reaction with CO. The peak is present when both CO and $O_2$ are present during reaction as shown in **d**. **e** APXPS for each of the reaction conditions in **a–d**. Data were acquired at beamline 9.3.2 of the ALS.

together with oxygen vacancies, is responsible for the enhanced catalytic reactivity of our Au(111)-supported nonstoichiometric $CoO_x$ monolayer films.

## DFT calculations of the reaction between CO and $CoO_x$/Au

To further understand the reactivity of $CoO_x$ structures on Au(111), DFT calculations of the reaction between CO and surface O were performed on $CoO_{x<1}$/Au(111), CoO/Au(111), and $CoO_{x>1}$/Au(111) using the calculated structural models in Fig. S7.

For $CoO_{x<1}$/Au, like on $CoO_{x<1}$/Pt[4], CO adsorbs exergonically by -0.35 eV atop a Co atom not bound to O, in agreement with the presence of molecular CO detected by APXPS (Fig. S5). The adsorbed CO can react with two types of surface O located at either a Face Centered Cubic site (FCC) or a Hexagonal Close Packed site (HCP) site (with respect to surface Co). The reaction between adsorbed CO and surface O is endothermic, by 1.35 eV for O at the FCC site and 1.17 eV for O at the HCP site, which agrees with the required heating of the $CoO_{x<1}$/Au film before reaction between adsorbed CO and surface O could take place. Next, the formation of carbonate groups on CoO/Au was investigated by DFT calculations. It was found to be unlikely for carbonate groups to form at the terraces of a stoichiometric bilayer CoO film, because the formation of $CO_2$ was calculated to be just as exothermic as that of $CO_3^{2-}$ (Fig. S4), indicating that, if formed, they should be unstable unless under a high $CO_2$ partial pressure. We tested the sensitivity of carbonate group formation to changes in Co-Co spacing by performing the same calculations over a CoO film with a narrower 3.00 Å Co-Co spacing, but the same trend was observed. Since the structure of bulk $CoCO_3$ is known[21], we examined the possibility that the formation of carbonates induced the restructuring of the oxide layer. The calculations indicate that a flat $CoCO_3$ overlayer on Au(111) is unstable and weakly bound. Instead, the optimized structures were found to dewet and relax away from the Au surface (Fig. S8). This is supported by the results in Fig. 4d where the dewetting of the oxide upon heating above 150 °C and beyond is revealed by the rapid increase in the Au4f XPS peak. Thus, it is likely that the experimentally observed carbonate groups are located on a restructured CoO terrace[7,22] or that they form at the edge of CoO islands which retain their 2D structure, as previously reported for CoO films on Pt(111)[4]. We note that the formation of carbonates on the CoO film with 3.00 Å Co-Co spacing also induced a dewetting reconstruction of interfacial Co, where the Co cation detaches from the Au substrate and moves above the surface-bound O (Fig. S4f). This restructuring also supports a more complex structural transformation of CoO upon the formation of carbonate groups.

Finally, to understand the superior reactivity of CO oxidation on $CoO_{x>1}$/Au, DFT calculations were performed to obtain the free energy

barriers of the CO-lattice O reaction (Figs. 7a and S9). Two possible structures for the $CoO_{x>1}$/Au were evaluated: a $Co_3O_4$ film (Fig. 7) and a $CoO_2$ film (Fig. S9). The structure of $CoO_x$ films supported on Au and Pt under oxidative conditions have been extensively characterized. For $CoO_x$/Au, it has been proposed that the O-rich $CoO_2$ phase is the most stable configuration under $O_2$[7,10,23]. Following our previous calculations of the structure of $Co_3O_4$(111) surfaces, a Co-poor and O-rich surface was chosen to simulate a $Co_3O_4$(111) film[24,25]. On this termination, only Co cations originally in bulk tetrahedral sites and O anions are exposed. To initiate the reaction, CO binds weakly on exposed Co. The adsorbed CO can react readily with O atoms bound to Co neighbors by crossing a 0.49 eV barrier. At 100 °C and 100 mTorr of CO, the net free energy barrier for CO-lattice -O reaction is 1.04 eV. We note that this barrier is lower than the CO-lattice O reaction energy over $CoO_{0.5}$ (1.17 eV, Fig. S5b), making $Co_3O_4$ more reactive. Further, under steady-state CO oxidation, the sub-oxidized phase was not observed. On the other hand, the $CoO_2$/Au(111) film appears less reactive than $Co_3O_4$ as a 1.24 eV initial barrier is required for the reaction between CO gas and lattice O (Fig. S9). Weakly bound $CO_2$ produced by this exothermic reaction can either desorb or further react with lattice O by crossing a 0.04 eV barrier (Fig. 7b). The formation of the O vacancy is linked to the reduction of subsurface Co. Upon the formation of the O vacancy, the magnetic moments of the two subsurface Co atoms surrounding the O vacancy shifted from 1.85 and 1.84 $\mu_B$ to 1.90 $\mu_B$, while the magnetic moment of the third subsurface Co changed from -0.09 to 2.59 $\mu_B$, indicating reduction. Note that the electronic energy of reaction is −1.80 eV relative to CO gas, which is much more exothermic than the reaction over CoO (Fig. S4). The $CO_3^{2-}$ group formed in this step is a bidentate species interacting with adjacent exposed Co. Although easy to form, the $CO_3^{2-}$ group can also decompose by crossing a 0.71 eV barrier (Fig. 7b). Carbonates formed by reaction between $CoO_2$ and the $CO_2$ product are also easy to decompose, requiring an even smaller barrier of 0.21 eV (Fig. S9). Under a low $CO_2$ partial pressure ($P_{CO2} = 0.1$ mTorr), the exergonic $CO_2$ desorption prevents the easily formed $CO_3^{2-}$ from poisoning the surface. Although the Au substrate does not participate in the CO oxidation reaction because of the weak adsorption of CO and inefficient dissociation of $O_2$, it can still modify the reactivity of the first $CoO_x$ layer in contact with the substrate. Our most recent results suggest that Au does indeed modify slightly the reactivity of the first monolayer CoO film, but has a lessened effect on the reactivity of the second layer, which remain slightly lower than that of the first.

In summary, the structure and reactions of Au-supported ultrathin $CoO_x$ catalyst films in the presence of $CO/O_2$ mixtures, were studied with the goal of determining the evolution of the catalyst structure and the nature of the active sites involved in CO oxidation. It

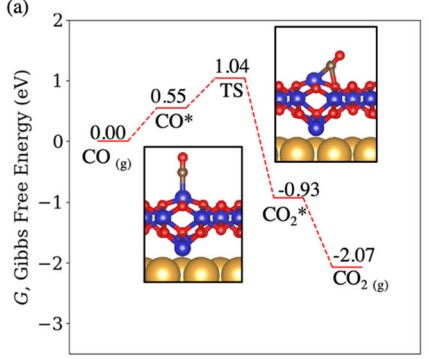
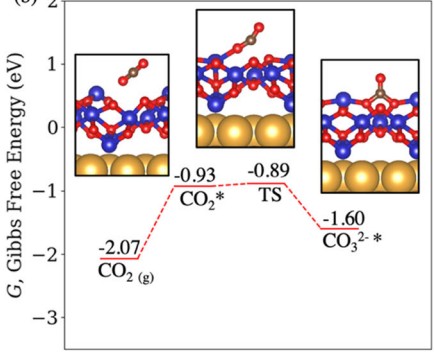
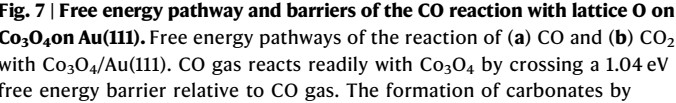

**Fig. 7 | Free energy pathway and barriers of the CO reaction with lattice O on $Co_3O_4$ on Au(111).** Free energy pathways of the reaction of (**a**) CO and (**b**) $CO_2$ with $Co_3O_4$/Au(111). CO gas reacts readily with $Co_3O_4$ by crossing a 1.04 eV free energy barrier relative to CO gas. The formation of carbonates by reaction between lattice O and $CO_2$ gas is unfavorable as carbonates will readily decompose (right-to-left in panel **b**). The free energies of gas CO and $CO_2$ were calculated at 373.15 K, $P_{CO} = 100$ mTorr, and $P_{CO2} = 0.1$ mTorr. Color scheme: Au: yellow; C: brown; Co: blue; O: red.

was found that $CoO_{x<1}$ films have a high chemical reactivity towards CO oxidation under oxygen-lean conditions at mild temperatures (from 100 °C to 150 °C), which we attribute to the presence of O vacancies near $Co^{\delta+}$ where CO can adsorb and react with neighboring lattice oxygen. In contrast, on stoichiometric CoO, the CO adsorbs forming carbonates that poison the CO oxidation and trigger restructuring and dewetting of the oxide film. The formation of carbonates, the decomposition, and dewetting of the $CoO_x$ to form 3D clusters divide the reaction into two regimes of high catalytic activity. One is characteristic of oxygen-lean conditions containing both $Co^0$ and $Co^{2+}$, the other is characteristic of oxygen-rich conditions containing $Co^{2+}$ and $Co^{3+}$ species. The $Co^{3+}$ can be reduced by CO to $Co^{2+}$ and regenerated by $O_2$ back to $Co^{3+}$ at temperatures of 100 °C and below. Using ResPES, we demonstrated the important role of $Co^{3+}$ as the most active catalyst site under reaction conditions. DFT calculations indicate that the high reactivity is due to a lower energy barrier for C-O bond formation and shows the pathway leading to restructuring and dewetting of the oxide film upon formation of carbonates. Our findings provide a general understanding of the enhanced catalytic reactivity of cobalt oxide catalysts and underline the double restructuring paradigm of chemical (i.e. Co oxidation state) and topographical ($CoO_x$ detachment and dewetting from the Au substrate) restructuring of the catalyst induced by the reactants composition and temperature.

## Methods

### Experimental methods

APXPS measurements were performed at beamline 9.3.2 of the Advanced Light Source (ALS) at the Lawrence Berkeley National Laboratory, at beamline 23-ID-2 (IOS) of the National Synchrotron Light Source II (NSLS-II) at Brookhaven National Laboratory, and at the Experimental Station 13-2 at the Stanford Synchrotron Radiation Light source (SSRL) at SLAC National Accelerator Laboratory. The Au(111) surface was cleaned by cycles of sputtering (5 min at $3 \times 10^{-5}$ Torr of $Ar^+$ at 1 keV energy), and annealing (10 min at 500 °C) until only Au was detected by XPS. Cobalt films were deposited on the clean Au(111) surface by evaporation from a Co rod (Goodfellow, 99.99 + %) using a SPECS e-beam evaporator. The amount of Co was measured by using XPS peak intensities, calibrated with a quartz crystal microbalance (QCM). The amount of Co on the surface is reported in monolayer equivalents (MLE), defined as the amount of Co that would form a complete monolayer if its wetting of Au was perfect. The reported MLE values have an estimated error bar of ±20% (see SI). The coverage of $Co^0$ for a given number of MLE was determined using the Co 3p to Au 4 f intensity ratio (Fig. S2) in comparison with simulated ratios from the quantitative photoelectron simulation package SESSA v2.2[26] (Fig. S3). After deposition, the films were oxidized by exposure to $O_2$ at room temperature (RT) or at 200 °C. To ensure high CO purity, the CO gas was passed through a carbonyl trap heated to ~240 °C before entering the measurement chamber. Total gas pressures were monitored with Baratron capacitance pressure gauges. Photon energies of 740, 475, 260, and 920 eV were used to generate photoelectrons with kinetic energies between 150 and 200 eV for the O 1 s, C 1 s, Co 3p, Co 2p, and Au 4f photoelectrons, respectively, which have mean free paths of ~ 5 Å. Reported binding energies are given with respect to the Fermi level.

### Computational methods

DFT calculations were performed using the Vienna Ab initio Simulation Package (VASP) version 5.4.1[27-29]. The exchange-correlation energy was calculated using the Perdew-Burke-Ernzerhof (PBE) functional[30]. Spin polarization was used in all calculations. The projector-augmented wave (PAW) method was used to describe the core electrons. The one-electron wavefunctions were expanded using a set of plane waves with kinetic energy up to 500 eV. To correct for the self-interaction error of Co 3d electrons, a Hubbard-like onsite repulsion term (DFT + $U$) was included in the calculations using Dudarev's approach[31], with an effective U value ($U_{eff}$) of 3.5 eV, which has been used in the literature to study the bulk and surface redox properties of $CoO_x$[32]. The CoO row-wise antiferromagnetic state was maintained in all calculations. More details regarding the spin state of Co cations are provided in the SI[4,33]. Structural relaxation for reaction intermediates was performed using the conjugate gradient algorithm. Transition states were first searched using the nudged elastic band (NEB) and climbing image (CI) NEB algorithms[34,35]. The highest energy image of each CI-NEB calculation was then refined using the Dimer and quasi-Newton algorithms[36]. The electronic structure in each self-consistent field (SCF) cycle was considered converged when the difference in total energy of consecutive steps fell below $10^{-6}$ eV. Atomic positions were considered converged when the Hellmann-Feynman forces on unconstrained atoms fall below 0.02 eV/Å. Free energies of CO and $CO_2$ gas were approximated using their translational and rotational partition functions[37]. The structure of ultrathin $CoO_x$ films on the hexagonal (111) surfaces of Au and Pt by STM has been studied extensively in the past[6,10,22,38]. In all cases the films form hexagonal lattices with slightly different unit cells from that of the metal substrate leading to formation of Moire patterns. Our previous STM and DFT studies of $CoO_x/Pt(111)$[4] and $FeO_x/Pt(111)$[39-46], and the more recent work of Zeuthen et al. on $FeO_x/Pd(111)$[47], again report similar structures. Here we follow the crystallographic structure of the $CoO_x/Au(111)$ films and their change during reaction by theoretical modeling using DFT. Details of the structural models are described in the SI (Fig. S7).

## Data availability

All data are available upon request.

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

## Acknowledgements

This work was supported by the Office of Basic Energy Sciences (BES),
Chemical Sciences, Geosciences, and Biosciences Division, of the U.S.
Department of Energy (DOE) under Contract DE-AC02-05CH11231, FWP
CH030201 (Catalysis Research Program). L. J. F. acknowledges support
from the Alexander von Humboldt Foundation, Bonn, Germany. It used
resources of the Advanced Light Source, a U.S. DOE Office of Science
User Facility under contract no. DE-AC02-05CH11231, the 23-ID-2 (IOS)
beamline of the National Synchrotron Light Source II, a User Facility
operated for the DOE Office of Science by Brookhaven National
Laboratory under Contract No. DE-SC0012704, and the Stanford Syn-
chrotron Radiation Light Source, SLAC National Accelerator Laboratory,
supported by the U.S. Department of Energy, Office of Science, Office of
Basic Energy Sciences under Contract No. DE-AC02-76SF00515. X.Z.
was supported by NSF-BSF grant number 1906014. The DFT calculations
in this work used computational and storage services associated with
the Hoffman2 cluster at the UCLA Institute for Digital Research and
Education (IDRE), and the Bridges-2 cluster at the Pittsburgh Super-
computing Center (supported by National Science Foundation award
number ACI-1928147) through the Extreme Science and Engineering
Discovery Environment (supported by National Science Foundation
grant number ACI-1548562) grant TG-CHE170060.

## Author contributions

M.S. oversaw the project. H.C., L.J.F., H.K., X.Z., and J.O.M. conduct the
APXPS experiment supported by the beamline scientists: S.N., A.H., I.W.,
and H.O. Data analysis were done by H.C. and M.S. Simulations were
carried out by G.Y. and P.S. The paper was written by H.C., G.Y., P.S., and
M.S., with contributions from all authors. A.T.B. and M.J. helped to
review and edit the manuscript.

## Competing interests

The authors declare no competing interests.
