## [Peer Review File · Nature Communications]

REVIEWER COMMENTS

Reviewer #1 (Remarks to the Author):

The promotion of CO oxidation by cobalt oxide is discussed in a number of publications, but a comprehensive picture has not yet been obtained. The submitted manuscript targets this topic, addressing especially the catalyst's stoichiometry and morphological/chemical transitions. This is a meaningful approach, but there are some remarks.

- The films are quite thin, in the monolayer range, which means that their properties are affected by the gold substrate. This will also be the case for the catalytic properties, which limits the scope of the results to quite thin cobalt oxide films on Au(111) with the gold being a relevant ingredient.
- A structural characterization with LEED, STM, etc was not performed. It may be the case, that the DFT optimized structure is correct (or at least near to the real structure), but an experimental verification would improve the level of trust.
- In the context of the SESSA simulation discussion it is said that cobalt and gold are immiscible. While this may be correct, it may still happen that thermodynamics drives cobalt atoms into the gold at elevated temperature (oxidation at 200°C). Thus, the real Co coverage may be smaller than the reported one.
- The ResPES experiments were performed to separate Co²⁺ and Co³⁺ contributions (some degree of differentiation should actually also be possible with the Co 2p levels if the data are measured with reasonably good statistics)

The resonances in ResPES have a certain width, which means that there will be a contribution from the other component at both resonance energies (Co²⁺ and Co³⁺). It appears that for the Co²⁺ case, off-resonance intensity is considered, while Co²⁺ intensity is considered for Co³⁺ (arrows in Fig. 5d). However, the contribution of Co³⁺ to the Co²⁺ spectrum is seemingly not considered. For the reader it may also be a bit surprising that there is such a high intensity at the Co³⁺ resonance in Fig. 5e. The resonance energies may also be somewhat different for CoO and Co₃O₄, likewise the linear correction factors. The authors may consider clarifying the description of the spectral Co²⁺/Co³⁺ separation a bit.

- Regarding the new paradigm: it appears that this refers to the combined changes of the morphology and the chemical state. Is this really so new? There have been operando TEM studies (Schlögl/Willinger but also others) where the authors report shape and chemical phase changes during a reaction. This also seems to involve both types of change.
- The manuscript does not contain catalytic reactivity studies. Thus, a reactivity comparison of the films is not really possible (beyond what follows from the spectroscopic data) although this could be interesting given that the manuscript deals with catalytic CO oxidation.

The results seem to sound. However, the limited scope of the data, the missing structural characterization and the missing reactivity data lower the relevance somewhat.

Reviewer #2 (Remarks to the Author):

This manuscript investigated the topographic restructuring and evolution of CoO_x catalysts supported on Au(111) single crystal surfaces in response to reaction conditions. The active sites for CO oxidation were determined by characterization methods and DFT calculations. I would conclude the work is of potential interest to be published on Nat. Commun. However, the following issues regarding the theoretical calculations need to be carefully clarified.

1. Since CoO is antiferromagnetic below Neel temperature, the DFT calculations need to specify the Co atomic magnetic moment, otherwise it is hard to achieve accurate results.
2. The authors do not provide a convincing explanation for the contradictory conclusions between the experiments and the DFT calculations of carbonate formation on the stoichiometric CoO films.
3. The authors proposed that Co³⁺ sites have a unique role in CO oxidation, but lacked a reasonable analysis to elucidate the chemical nature. Also, the authors considered that partially oxidized films (CoO_x<1) containing CoO are efficient catalysts, so a comparison of these two sites is necessary.
4. There have been many similar studies on the structural evolution of CoO_x catalysts during CO oxidation, and the authors should compare this work with other studies.

Reviewer #3 (Remarks to the Author):

The manuscript by Chen et al. (NCOMMS-23-25699-T) reports the existence of three different CO oxidation reaction regimes which depend on the chemical state of the catalyst which, in turn, depends on the gas phase CO/O₂ stoichiometry. The authors employed Ambient Pressure XPS (APXPS) and Resonant Photoemission Spectroscopy (ResPES) to monitor the oxidation state of model cobalt catalyst under reaction conditions. The key result of this study is the observation of Co³⁺ ions associated with the formation of Co₃O₄ phase and their involvement in CO oxidation. Note that detection of small amounts of Co³⁺ based on the core level spectra is extremely difficult due to its complex shape. The reported results are particularly important for cobalt-based catalysts often used in combination with supported noble metal nanoparticles, where redox interactions play a crucial role in the catalyst reactivity and selectivity. Monitoring and quantitative analysis of the oxidation

state of cobalt-based catalyst allows to obtain comprehensive insights into catalyst activity which is controlled by redox interactions.

The text of the manuscript is clearly written. The literature review is comprehensive. The experimental data are of good quality and should be easily reproduced. The data were analyzed and interpreted carefully and are presented in sufficient detail. However, I have concerns about the calibration of RER parameter (see questions below). The experimental evidence for the conclusions is strong. (The evaluation of theoretical study is out of my expertise).

I recommend to accept the manuscript for publication in Nature Communications after the authors address the question listed below.

1) Figure 3. Authors should comment on the appearance of sharp peak in C 1s region (around 283.0 eV) obtained from 1 ML CoO_{0.25} catalyst under exposure to CO at and above 100 C. This could point to formation of cobalt carbides due to additional reaction pathway, e.g. via CO disproportionation.

2) Figure 4. The signal in the Co 2p region is unusually low at 375 C under exposure to CO. The authors explain this by combined oxide reduction upon decomposition of carbonates and CoO dewetting and formation of CoO clusters. Did the authors verified such a scenario by simulation in SESSA? Can the authors rule out desorption of cobalt carbonyl species?

3) Lines 283-294. The authors determined RER of stoichiometric Co₃O₄ to be 0.67. This value is very different to the value determined earlier on well-ordered stoichiometric Co₃O₄(111) films (RER=0.9) in Ref. 27. It is hard to believe that stoichiometric compound could give such different values. The authors should provide evidence that their Co₃O₄ sample used for calibrations of ResPES has a structure and stoichiometry of Co₃O₄.

Minor

a) Lines 203. Check the labeling of panels in Figure 3. O 1s and Co 2p are shown in (c) and (a), respectively

Response to the Reviewer 1:

Reviewer #1 (Remarks to the Author):

The promotion of CO oxidation by cobalt oxide is discussed in a number of publications, but a comprehensive picture has not yet been obtained. The submitted manuscript targets this topic, addressing especially the catalyst's stoichiometry and morphological/chemical transitions. This is a meaningful approach, but there are some remarks.

Reply: We thank the reviewer for careful reading of the manuscript and for raising the constructive comments. Our point-by-point responses are listed below.

• The films are quite thin, in the monolayer range, which means that their properties are affected by the gold substrate. This will also be the case for the catalytic properties, which limits the scope of the results to quite thin cobalt oxide films on Au(111) with the gold being a relevant ingredient.

Reply: The reviewer raises a good and important point. The Au(111) was chosen as substrate for the growth of oxide films because, unlike Pt[4, 5] or other noble metals[6-8] that are active in the CO oxidation reaction, the Au (111) surface does not participate in the CO oxidation reaction because of its weak adsorption of CO and because it cannot dissociate O₂ efficiently. However, as the reviewer points out, Au can modify the reactivity of the monolayer CoO film. We studied this question with the help of DFT calculations and found that the reaction energy of the CO oxidation reaction on a second CoO layer is higher than that of the first layer by ~0.4 eV. This topic, together with a comparison between Au and Pt substrates is discussed in detail in a paper now in preparation. Here the focus is on the reactivity of different Co oxidation states and on the configurational changes that occur in response to the reactant gas composition. Following the reviewer's comment, we have included a mention of the influence of Au in the revised version of the manuscript.

Proposed changes (highlighted in yellow):

Page 16 and line 422 of the **Main text**:

“Although the Au substrate does not participate in the CO oxidation reaction because of the weak adsorption of CO and inefficient dissociation of O₂, it can still modify the reactivity of the first CoO_x layer in contact with the substrate. Our most recent results suggest that Au does indeed modify slightly the reactivity of the single monolayer CoO film, but has a lessened effect on the reactivity of the second layer, which remain slightly lower than that of the first.”

• A structural characterization with LEED, STM, etc was not performed. It may be the case, that the DFT optimized structure is correct (or at least near to the real structure), but an experimental verification would improve the level of trust.

Reply: We agree with the reviewer. The Au-CoO_x system has been studied using LEED, STM and other surface science techniques by the groups of Prof. Lauritsen in Denmark[9-12] and Prof. E. I. Altman in the United States[13], whose results on the structure of the oxide layers are in line with our DFT calculations. In our previous STM results with similar systems, CoO_x/Pt(111)[5], and FeO_x/Pt(111) [14-20], all prepared in similar conditions, we found similar structures, also in line with our DFT calculations. This makes us confident in the calculated DFT structures of CoO_x/Au(111). We have included the following in the revised version of the manuscript.

Proposed changes:

Page 5 and line 121 of the **Main text**:

“partition functions. [21] ... The structure of ultrathin CoO_x films on the hexagonal (111) surfaces of Au and Pt by STM has been studied extensively in the past.[9-12] In all cases the films form hexagonal lattices with slightly different unit cells from that of the metal substrate leading to formation of Moire patterns. Our previous STM and DFT studies of CoO_x/Pt(111)[5] and FeO_x/Pt(111) [14-20, 22], and the more recent work of Zeuthen et al. on FeO_x/Pd(111)[23], again report similar structures. Here we follow the crystallographic structure of the CoO_x/Au(111) films and their change during reaction by theoretical modeling using the same DFT approach. Details of the structural models are described in the **SI (Fig. S3)**.

• In the context of the SESSA simulation discussion it is said that cobalt and gold are immiscible. While this may be correct, it may still happen that thermodynamics drives cobalt atoms into the

gold at elevated temperature (oxidation at 200°C). Thus, the real Co coverage may be smaller than the reported one.

Reply: As stated in the manuscript, heating under reducing conditions accelerates the dewetting of the CoO_x , which is related to the weak affinity for formation of Co-Au bonds. The immiscibility of Co and Au in the temperature conditions of our experiment was also manifested in our unsuccessful attempts to synthesize Au-Co nanoparticle alloys by intimately mixing AuCl_3 and $\text{Co}(\text{acac})_2$ molecular precursors. This is strong evidence of the difficulty to form a stable Co-Au alloy under our conditions. The immiscibility between Au and Co is also in line with the Au-Co bimetallic phase diagram[24], that exhibits no significant Co fraction in Au for temperatures below 500 °C.

• The ResPES experiments were performed to separate Co^{2+} and Co^{3+} contributions (some degree of differentiation should actually also be possible with the Co 2p levels if the data are measured with reasonably good statistics)

Reply: Yes, the reviewer is right. Both we [5] and others [25-27] have fitted the Co XPS peaks that are composed of overlapping Co peaks in various oxidation states (0, +1, +2, and +3) and could obtain a good fit with the experimental peaks by convolution of these components. The fits provided a possible answer to the amount of each Co^{n+} species. However, we believe that this is not sufficiently convincing. After all, with sufficient peaks one can fit anything. ResPES, on the other hand, separates unambiguously these oxidation states.

• The resonances in ResPES have a certain width, which means that there will be a contribution from the other component at both resonance energies (Co^{2+} and Co^{3+}). It appears that for the Co^{2+} case, off-resonance intensity is considered, while Co^{2+} intensity is considered for Co^{3+} (arrows in Fig. 5d). However, the contribution of Co^{3+} to the Co^{2+} spectrum is seemingly not considered. For the reader it may also be a bit surprising that there is such a high intensity at the Co^{3+} resonance in Fig. 5e. The resonance energies may also be somewhat different for CoO and Co_3O_4 , likewise the linear correction factors. The authors may consider clarifying the description of the spectral $\text{Co}^{2+}/\text{Co}^{3+}$ separation a bit.

Reply: The reference oxides used here are CoO with only 2+, and stoichiometric Co₃O₄ with a mixture of 2+ and 3+. The fact that both the green and the blue spectra (new version of **Fig.5** shown in below **Fig.R1**) acquired at the two different resonant energies in **Fig.5d** have very similar shapes is because both originate from the same Co²⁺ state. The lower intensity of blue curve is due to the slightly off resonance value of the 781.2 eV photon energy used. It also tells us that there is no, or very little, Co³⁺ in this sample. So, there is nothing to be subtracted from the spectra. The fact that we used the color and photon energy for the Co³⁺ resonance in the previous version is what confused the reviewer. That is now corrected in the edited version of the manuscript (**Fig.R1**). In contrast, on Co₃O₄ (**Fig.5e**) we have two distinct lines for the spectra acquired at the two resonant energies, one for the 3+ state with the prominent peak in the red curve near 1 eV BE. The other spectrum (green line) corresponds to Co²⁺, and has a similar shape as that from the CoO sample in **Fig.5d**, as expected. We thank the reviewer for revealing our poor choice of colors and explanation. We have corrected this and also modified the figure to avoid this confusion.

Figure.R1 New version of the Fig.5 in the manuscript

To determine the linear correction factor (y), the $D(\text{Co}^{2+})$ and $D(\text{Co}^{3+})$ values from the Co₃O₄ film are needed. These values correspond to the length of the arrows at 5 eV BE and at 1 eV BE in **Fig.5e**. In this manner, the contribution of the d levels from Au the substrate is subtracted to get the $D(\text{Co}^{2+})$, while the contribution of both Co²⁺ and Au are subtracted in the arrow at 1 eV BE to get $D(\text{Co}^{3+})$.

Proposed changes:

To Page 11 and line 283 of the **Main text**:

“...photoemission states from 1 MLE of CoO (b), and of 1 MLE of Co₃O₄ (c) on Au(111). (X=Binding Energy, Y=exciting photon energy, Z= color coded photoemission intensity). (d) VB spectra of CoO for $\hbar\omega = 779.8$ eV, the resonant photon energy for the Co²⁺ (green trace), for $\hbar\omega = 781.2$ eV, slightly off resonance for the same state (blue trace); and for $\hbar\omega = 772$ eV (bottom light blue trace), dominated by the d-levels of the Au substrate. (e) VB spectra of Co₃O₄ for $\hbar\omega = 781.2$ eV, resonant photon energy for the Co³⁺ state (red trace); for $\hbar\omega = 779.8$ eV, resonant energy for the Co²⁺ (green trace); and for $\hbar\omega = 772$ eV off-resonance (bottom light blue), dominated by the d-levels of the Au substrate.

To Page 12 and line 295 of the **Main text**:

“...**Fig.5b-c**. For the CoO, only Co²⁺ species are present, with a resonant photon energy of $\hbar\omega = 779.8$ eV, as shown in the heat map. The VB d-states of Co²⁺ with peaks at ~ 5 eV and ~ 10 eV, are strongly enhanced at this photon energy (green trace in (Fig.5d)). For $\hbar\omega = 781.2$ eV, slightly off resonance, the spectrum is similar as expected, but less intense (blue trace), and for $\hbar\omega = 772$ eV (far from resonance), the VB spectrum (light blue trace) is dominated by the Au substrate. For Co₃O₄ (**Fig.5c**) the resonant photon energy for the Co³⁺ site is 781.2 eV, as shown by the maximum in the heat map. The VB spectrum at this photon energy shows several Co³⁺ d-band peaks: a sharp one at 1.0 eV, and others around 10 eV, and 5.0 eV (**Fig.5e**). The VB...”

• *Regarding the new paradigm: it appears that this refers to the combined changes of the morphology and the chemical state. Is this really so new? There have been operando TEM studies (Schlögl/Willinger but also others) where the authors report shape and chemical phase changes during a reaction. This also seems to involve both types of change.*

Reply: The reviewer is correct that changes in catalyst structure driven by reactants and products is not a novelty, as we have also shown in the past. However, old beliefs are persistent and it is

still normally assumed that catalysts do not change. Here we add an example of a double restructuring: chemical and structural, both strongly correlated, driven by reactants and products.

• The manuscript does not contain catalytic reactivity studies. Thus, a reactivity comparison of the films is not really possible (beyond what follows from the spectroscopic data) although this could be interesting given that the manuscript deals with catalytic CO oxidation.

Reply: Correct. We can only conclude that the activity increases by the presence of Co^{3+} with the novelty that this can now be unambiguously related to these oxidized Co species, as proven by our in operando spectroscopic results.

• The results seem to sound. However, the limited scope of the data, the missing structural characterization and the missing reactivity data lower the relevance somewhat.

Reply: We really appreciated this positive comment. Although, suffering from limited access to the in-situ techniques, such as STM and other surface sensitive methods, which can provide more operando observations about CoO_x -Au model catalyst during the CO oxidation reaction, we successfully followed changes of the chemical states and morphological changes of CoO_x by APXPS, which we further supported by advanced DFT calculations.

Reviewer #2 (Remarks to the Author):

• This manuscript investigated the topographic restructuring and evolution of CoO_x catalysts supported on Au(111) single crystal surfaces in response to reaction conditions. The active sites for CO oxidation were determined by characterization methods and DFT calculations. I would conclude the work is of potential interest to be published on Nat. Commun. However, the following issues regarding the theoretical calculations need to be carefully clarified.

Reply: We thank the reviewer for his/her careful reading of the manuscript and for raising constructive comments. Our point-by-point responses are listed below.

• *Since CoO is antiferromagnetic below Neel temperature, the DFT calculations need to specify the Co atomic magnetic moment, otherwise it is hard to achieve accurate results.*

Reply: Indeed, we have accounted for the spin state of Co and its antiferromagnetism in our calculations. In the updated manuscript and SI, we have added details of the spin states of the Co cations and an extra line in the methods section of the manuscript referring to them.

Proposed changes:

To Page 5 and line 110 of the **Main text**:

“... surface redox properties of CoO_x. [10] CoO's row-wise antiferromagnetic state was maintained in all calculations; more details regarding the spin state of Co cations are provided in the SI. Structural relaxation...”

To Page 5 of the **SI**:

Line 104: “...allowed to relax. Following our previous work on CoO_{x<1}/Pt, a row-wise antiferromagnetic structure was used for all CoO_{x<1} models. In order from top-left to bottom-right in the outlined unit cell (Fig. S3a), the magnetic moments of the Co atoms were found to be -2.32, -1.99, 2.49, and 2.49 μ_B . For these structures...”

Line 115: “... Following previous studies of thin CoO films, a row-wise antiferromagnetic structure was used for all CoO models. The absolute magnetic moment of the Co cations in the films with 3.00 Å and 3.25 Å lattice spacing were found to be 2.36 and 2.57 $|\mu_B|$, respectively. For these structures...”

Line 132: “For Co₃O₄, the antiferromagnetic alignment of Co²⁺ cations in tetrahedral sites was maintained in all calculations. The magnetic moment of the outermost Co cation was found to be -2.85 μ_B , while the magnetic moments of the three Co cations in the subsurface were found to be -0.09, 1.85, and 1.84 μ_B . For CoO₂, in the order of top-left to bottom-right in the outlined unit cell (Fig. S3d), the magnetic moments of the Co atoms were found to be 0.00, 0.00, 1.26, and 1.26 μ_B .”

2. The authors do not provide a convincing explanation for the contradictory conclusions between the experiments and the DFT calculations of carbonate formation on the stoichiometric CoO films.

Reply: Although we were unable to find thermodynamically stable configurations of carbonate groups on the terraces of CoO films, the DFT-optimized geometry of both a single bidentate carbonate and the CoCO₃ film suggest that carbonates are metastable and induce restructuring and dewetting of the oxide film. A more complete study of this process would require much larger models. Beyond the formation of an unstable CoCO₃, we found that formation of carbonates on the CoO/Au(111) film with a 3.00 Å Co-Co spacing is more exothermic than that over the film with a 3.25 Å Co-Co spacing and induces a dewetting restructuring of CoO, where one interfacial Co has moved above O. We have added additional discussion to the description of the CoO reactivity.

Proposed changes:

Page 15 of the manuscript:

Line 375: "... (Fig. S7), indicating that, if formed, they should be unstable unless under a high CO₂ partial pressure."

Line 387: "...reported for CoO films on Pt(111). We note that the formation of carbonates on the CoO film with 3.00 Å Co-Co spacing also induced a dewetting reconstruction of interfacial Co, where Co detaches from the Au substrate and moves above the surface-bound O (Fig. S7f). This restructuring also supports a more complex structural transformation of CoO upon the formation of carbonate groups."

3. The authors proposed that Co³⁺ sites have a unique role in CO oxidation, but lacked a reasonable analysis to elucidate the chemical nature. Also, the authors considered that partially oxidized films (CoO_x<1) containing Co⁰ are efficient catalysts, so a comparison of these two sites is necessary.

Reply: We believe that the over-oxidized Co³⁺ species are responsible for the reactivity as they are readily reduced in the reaction between CO and surface O, which is reflected in the change of the Co cations' magnetic moments after reaction. The reduction of Co²⁺ on the other hand is much more difficult.

In addition, although the sub-oxidized films are also somewhat reactive towards CO, they are unlikely to be responsible for the reactivity of CoO_x/Au as the phase was not observed in O-rich CO oxidation environments and is unlikely to be a contributor to reactivity. We have added both explanations to the manuscript.

Proposed changes:

To Page 16 of the manuscript:

Line 406: "...more reactive. Further, under steady state CO oxidation, the sub oxidized phase was not observed."

Line 410: "... (Fig. 7b). The formation of the O vacancy is linked to the reduction of subsurface Co. Upon the formation of the O vacancy, the magnetic moments of the two subsurface Co surrounding the O vacancy shifted from 1.85 and 1.84 μ_B to 1.90 μ_B, while the magnetic moment of the third Co changed from -0.09 to 2.59 μ_B, indicating reduction. Note that the electronic energy of reaction is -1.80 eV relative to CO gas, which is much more exothermic than the reaction over CoO (Fig. S7). The CO₃²⁻ group..."

4. There have been many similar studies on the structural evolution of CoO_x catalysts during CO oxidation, and the authors should compare this work with other studies.

Reply: Indeed, the stability of the CoO_x phases have been studied extensively by the group of Lauritsen, where they proposed that the most stable phase is CoO₂ under an oxidative environment. We have updated the text to refer to their findings and support our choice of highly oxidized phases.

Proposed changes:

Page 15 of the manuscript:

Line 395: "... a CoO₂ film (Fig. S9). The structure of CoO_x films supported on Au and Pt under oxidative conditions have been extensively characterized. For CoO_x/Au, it has been proposed that the O-rich CoO₂ phase is the most stable configuration under O₂. [11, 28, 29] Following our..."

Reviewer #3 (Remarks to the Author):

The manuscript by Chen et al. (NCOMMS-23-25699-T) reports the existence of three different CO oxidation reaction regimes which depend on the chemical state of the catalyst which, in turn, depends on the gas phase CO/O₂ stoichiometry. The authors employed Ambient Pressure XPS (APXPS) and Resonant Photoemission Spectroscopy (ResPES) to monitor the oxidation state of model cobalt catalyst under reaction conditions. The key result of this study is the observation of Co³⁺ ions associated with the formation of Co₃O₄ phase and their involvement in CO oxidation. Note that detection of small amounts of Co³⁺ based on the core level spectra is extremely difficult due to its complex shape. The reported results are particularly important for cobalt-based catalysts often used in combination with supported noble metal nanoparticles, where redox interactions play a crucial role in the catalyst reactivity and selectivity. Monitoring and quantitative analysis of the oxidation state of cobalt-based catalyst allows to obtain comprehensive insights into catalyst activity which is controlled by redox interactions.

The text of the manuscript is clearly written. The literature review is comprehensive. The experimental data are of good quality and should be easily reproduced. The data were analyzed and interpreted carefully and are presented in sufficient detail. However, I have concerns about the calibration of RER parameter (see questions below). The experimental evidence for the conclusions is strong. (The evaluation of theoretical study is out of my expertise).

Reply: We really thank the reviewer for positive comments and very important questions. Our point-by-point responses are listed below.

I recommend to accept the manuscript for publication in Nature Communications after the authors address the question listed below.

1) Figure 3. Authors should comment on the appearance of sharp peak in C 1s region (around 283.0 eV) obtained from 1 ML CoO_{0.25} catalyst under exposure to CO at and above 100 C. This could point to formation of cobalt carbides due to additional reaction pathway, e.g. via CO disproportionation.

Reply: We agree with the reviewer's comment that the sharp peak at 283.0 eV may originate from the CO disproportionation on the Co^0 sites at elevated temperature. However, this CO disproportionation reaction ($2\text{CO} \rightarrow \text{C} + \text{CO}_2$) would not contribute to the decrease of lattice oxygen (**Fig.3c, O1s spectra**) when the reaction temperature raised from 100 °C to 150 °C. Therefore, the reaction between CO and lattice oxygen of $\text{CoO}_{0.25}$ is favorable at this temperature. Following the reviewer's comment, we have included a mention of this 283.eV peak in the revised version of the manuscript.

Proposed changes:

To Page 9 and line 216 of the **Main text**:

“The new small peak at 283.0 eV originates from cobalt carbides(CoC_x), suggesting CO dissociation at Co^0 sites at elevated temperature.[30] Raising the ...”

2) Figure 4. The signal in the Co 2p region is unusually low at 375 C under exposure to CO. The authors explain this by combined oxide reduction upon decomposition of carbonates and CoO dewetting and formation of CoO clusters. Did the authors verified such a scenario by simulation in SESSA? Can the authors rule out desorption of cobalt carbonyl species?

Reply: This reviewer raises a good point. We verify the dewetting process of reduced CoO_x clusters and rule out the formation of the carbonyl species as the reappearance of Co2p peak after re-oxidizing this reduced CoO_x clusters on Au (111) under 100 mTorr of O_2 at ~350 °C (**Fig.R2**). The explanation of this dewetting process is to be found in the phase diagram of the Au-Co binary system[24] which, as mentioned before, shows that Co and Au are immiscible. Thus, reduction of Co^{2+} to Co^0 leads to dewetting by diffusion of the Co^0 and formation of 3-D Co clusters. A similar behavior has been reported by Parkinson et al. [31] showing that well-dispersed Ir adatoms on a magnetite surface merged into large clusters after thermal annealing at higher temperature.

Figure.R2 Co2p XP spectra of ~1.0 MLE reduced CoO_x film under 100mTorr O₂ at ~350 °C.

Proposed changes:

Page 10 of the manuscript:

Line 243: "... from CoO_{x=1} to CoO_{x<1}.) Introduction of 100 mTorr of O₂ at ~350 °C caused the reappearance of the Co2p peak (see SI, Fig. S5). We discuss this reaction mechanism further in the theoretical **Section 3.6**"

To Page 6 of the SI:

Line 156: **Figure S5:** (a) Co2p XP spectra of 1.0 MLE reduced CoO_x film under 100mTorr O₂ at ~350 °C; ResPES of 1.0MLE CoO_x under (b)100mtorr O₂ and (c) 100mtorr CO at RT, respectively.

3) Lines 283-294. The authors determined RER of stoichiometric Co₃O₄ to be 0.67. This value is very different to the value determined earlier on well-ordered stoichiometric Co₃O₄ (111) films (RER=0.9) in Ref. 27. It is hard to believe that stoichiometric compound could give such different values. The authors should provide evidence that their Co₃O₄ sample used for calibrations of ResPES has a structure and stoichiometry of Co₃O₄.

Reply: We thank the reviewer for pointing this out. When we preparing this draft, our group members also discussed on this discrepancy. We did the ResPES on the reference samples, which were confirmed to be CoO and Co₃O₄ by the Co2p XP spectra. The difference in RER may

originate from the Au substrate's contribution. The thickness of the CoO_x film supported on the Ir(100) is ~ 6.0 nm (data from **Ref. 27**) while the thickness of the CoO_x film supported on the Au(111) is ~ 0.3 nm in this manuscript. Therefore, we can still observe some peaks on the off-resonance Valence Band spectra that correspond to d-levels of the Au substrate (**Fig.R3a**). However, the contributions from the substrate's d-levels in **Ref.27** is negligible (**Fig.R3b**). When we did the calculation of RER value on the nonstoichiometric CoO_x ultrathin layer on Au, we consistently subtracted the contributions from the Au d-levels. Therefore, the stoichiometry of CoO_x we get under the CO oxidation reaction is valid.

Figure.R3 ResPES spectra of (a) ~ 0.3 nm thick $\text{CoO}_x/\text{Au}(111)$ from our draft and (b) ~ 6.0 nm thick $\text{CoO}_x/\text{Ir}(100)$ from **Ref.27**. Note that, the color representing different Resonance spectra in above two panels.

Proposed changes:

Page 12 of the manuscript:

Line 313: "... Co_3O_4 is 0.67. Note that the contribution of the d levels from the Au substrate is subtracted to get the $D(\text{Co}^{2+})$, while the contribution of both Co^{2+} and Au are subtracted at ~ 1 eV BE to get $D(\text{Co}^{3+})$, as respectively indicated by two vertical arrows in **Fig. 5d**. Since the..."

Minor

a) Lines 203. Check the labeling of panels in Figure 3. O 1s and Co 2p are shown in (c) and (a), respectively

Reply: We thanks for the reviewer's reading carefully on our manuscript.

Proposed changes:

Page 8-9 of the manuscript:

Line 212: "... at the higher temperature) (Fig.3b). This is ..."

Line 213: "...Co⁰ peak at 778 eV (Fig. 3(b,c))" changes to "Co⁰ peak at 778 eV (Fig. 3a,3c)"

Line 223: "...and (b) O 1s core level regions" changes to "...and (c) O 1s core level regions"

References

1. Ning, Y., et al., *Nature of Interface Confinement Effect in Oxide/Metal Catalysts*. The Journal of Physical Chemistry C, 2015. **119**(49): p. 27556-27561.
2. Chen, H., et al., *CO and H₂ Activation over g-ZnO Layers and w-ZnO(0001)*. ACS Catalysis, 2018. **9**(2): p. 1373-1382.
3. Zhao, X., et al., *Growth of Ordered ZnO Structures on Au(111) and Cu(111)*. Acta Physico-Chimica Sinica, 2018. **34**(12): p. 1373-1380.
4. An, K., et al., *Enhanced CO oxidation rates at the interface of mesoporous oxides and Pt nanoparticles*. J Am Chem Soc, 2013. **135**(44): p. 16689-96.
5. Kersell, H., et al., *CO Oxidation Mechanisms on CoOx-Pt Thin Films*. J Am Chem Soc, 2020. **142**(18): p. 8312-8322.
6. Gao, F., et al., *CO Oxidation on Pt-Group Metals from Ultrahigh Vacuum to Near Atmospheric Pressures. I. Rhodium*. Journal of Physical Chemistry C, 2009. **113**(1): p. 182-192.
7. Gao, F. and D.W. Goodman, *CO oxidation over ruthenium: identification of the catalytically active phases at near-atmospheric pressures*. Physical Chemistry Chemical Physics, 2012. **14**(19): p. 6688-6697.
8. Gao, F., et al., *CO oxidation trends on Pt-group metals from ultrahigh vacuum to near atmospheric pressures: A combined in situ PM-IRAS and reaction kinetics study*. Surface Science, 2009. **603**(1): p. 65-70.
9. Morgenstern, K., et al., *Cobalt growth on two related close-packed noble metal surfaces*. Surface Science, 2007. **601**(9): p. 1967-1972.
10. Rattigan, E., et al., *The cobalt oxidation state in preferential CO oxidation on CoOx/Pt(111) investigated by operando X-ray photoemission spectroscopy*. Phys Chem Chem Phys, 2022. **24**(16): p. 9236-9246.
11. Walton, A.S., et al., *Interface controlled oxidation states in layered cobalt oxide nanoislands on gold*. ACS Nano, 2015. **9**(3): p. 2445-53.

12. Rattigan, E., et al., *Dewetting Transition of CoO/Pt(111) in CO Oxidation Conditions Observed In Situ by Ambient Pressure STM and XPS*. The Journal of Physical Chemistry C, 2023. **127**(18): p. 8547-8556.
13. Li, M. and E.I. Altman, *Shape, Morphology, and Phase Transitions during Co Oxide Growth on Au(111)*. The Journal of Physical Chemistry C, 2014. **118**(24): p. 12706-12716.
14. Galloway, H.C., J.J. Benítez, and M. Salmeron, *The structure of monolayer films of FeO on Pt(111)*. Surface Science, 1993. **298**(1): p. 127-133.
15. Galloway, H.C., J.J. Benítez, and M. Salmeron, *Growth of FeOx on Pt(111) studied by scanning tunneling microscopy*. Journal of Vacuum Science & Technology A: Vacuum, Surfaces, and Films, 1994. **12**(4): p. 2302-2307.
16. Galloway, H.C., P. Sautet, and M. Salmeron, *Structure and contrast in scanning tunneling microscopy of oxides: FeO monolayer on Pt(111)*. Phys Rev B Condens Matter, 1996. **54**(16): p. R11145-R11148.
17. Kim, Y.J., et al., *Interlayer interactions in epitaxial oxide growth: FeO on Pt(111)*. Physical Review B, 1997. **55**(20): p. R13448-R13451.
18. Merte, L.R., et al., *CO-induced embedding of Pt adatoms in a partially reduced FeO(x) film on Pt(111)*. J Am Chem Soc, 2011. **133**(28): p. 10692-5.
19. Liu, Y., et al., *Structure and Electronic Properties of Interface-Confined Oxide Nanostructures*. ACS Nano, 2017. **11**(11): p. 11449-11458.
20. Chen, H., et al., *Active Phase of FeOx/Pt Catalysts in Low-Temperature CO Oxidation and Preferential Oxidation of CO Reaction*. The Journal of Physical Chemistry C, 2017. **121**(19): p. 10398-10405.
21. Hill, T.L., *An introduction to statistical thermodynamics*. 1986: Courier Corporation.
22. Liu, Y., et al., *Enhanced oxidation resistance of active nanostructures via dynamic size effect*. Nat Commun, 2017. **8**: p. 14459.
23. Zeuthen, H., et al., *Structure of Stoichiometric and Oxygen-Rich Ultrathin FeO(111) Films Grown on Pd(111)*. The Journal of Physical Chemistry C, 2013. **117**(29): p. 15155-15163.
24. Okamoto, H., et al., *The Au-Co (Gold-Cobalt) system*. Bulletin of Alloy Phase Diagrams, 1985. **6**(5): p. 449-454.
25. Lykhach, Y., et al., *Quantitative Analysis of the Oxidation State of Cobalt Oxides by Resonant Photoemission Spectroscopy*. J Phys Chem Lett, 2019. **10**(20): p. 6129-6136.
26. Biesinger, M.C., et al., *Resolving surface chemical states in XPS analysis of first row transition metals, oxides and hydroxides: Cr, Mn, Fe, Co and Ni*. Applied Surface Science, 2011. **257**(7): p. 2717-2730.
27. Favaro, M., et al., *Understanding the Oxygen Evolution Reaction Mechanism on CoO(x) using Operando Ambient-Pressure X-ray Photoelectron Spectroscopy*. J Am Chem Soc, 2017. **139**(26): p. 8960-8970.
28. Fester, J., et al., *Comparative Analysis of Cobalt Oxide Nanoisland Stability and Edge Structures on Three Related Noble Metal Surfaces: Au(111), Pt(111) and Ag(111)*. Topics in Catalysis, 2016. **60**(6-7): p. 503-512.
29. Fester, J., et al., *Phase Transitions of Cobalt Oxide Bilayers on Au(111) and Pt(111): The Role of Edge Sites and Substrate Interactions*. J Phys Chem B, 2018. **122**(2): p. 561-571.

30. Wu, C.H., et al., *Ambient-Pressure X-ray Photoelectron Spectroscopy Study of Cobalt Foil Model Catalyst under CO, H₂, and Their Mixtures*. ACS Catalysis, 2017. **7**(2): p. 1150-1157.
31. Jakub, Z., et al., *Local Structure and Coordination Define Adsorption in a Model Ir₁/Fe₃O₄ Single-Atom Catalyst*. Angew Chem Int Ed Engl, 2019. **58**(39): p. 13961-13968.

REVIEWERS' COMMENTS

Reviewer #1 (Remarks to the Author):

The authors have mostly addressed the comments properly. However, questions regarding Figure 5 and the equation on line 312 remain. One can challenge the equation by applying it to defined systems where the $\text{Co}^{2+}/\text{Co}^{3+}$ concentration ratio is known, such as CoO with the spectra in Figure 5d. For these spectra, one gets $D(\text{Co}^{3+}) = \text{small, negative}$ and $D(\text{Co}^{2+}) = \text{comparably large}$. That is, $\text{RER} = D(\text{Co}^{3+})/D(\text{Co}^{2+}) = \text{somewhat small, negative}$. With that one gets $N(\text{Co}^{3+})/N(\text{Co}^{2+}) = 2.99 * \text{RER} = \text{also small, negative}$. It is not zero as it should be. If one considers a hypothetical system with only Co^{3+} , then one may get a similar set of spectra as those in Figure 5d, with the Co^{2+} resonance spectrum replaced by the Co^{3+} resonance spectrum and the intensity ratio being inverted (i.e, the spectrum at the Co^{3+} resonance energy would be more intense than the spectrum at the Co^{2+} resonance energy). One would have $D(\text{Co}^{3+}) = \text{small}$ and $D(\text{Co}^{2+}) = \text{comparably large}$ and $\text{RER} = \text{small}/\text{comparably large} = \text{somewhat small}$. That is, $N(\text{Co}^{3+})/N(\text{Co}^{2+}) = 2.99 * \text{somewhat small}$. However, it should be infinite. This seems to indicate that the equation is somewhat faulty.

Two details: 'Res' in Fig. 5e should probably be 'RER' and the colors of the labels in figs. 5d,e differ somewhat from the line colors (it is assumed that the authors wanted to have them identical).

Reviewer #2 (Remarks to the Author):

The authors have well addressed my comments. I recommend the publication of the manuscript on Nat. Commun.

Reviewer #3 (Remarks to the Author):

The authors properly addressed all questions. I recommend to accept the manuscript for publication in Nature Communications.

Lawrence Berkeley National Laboratory
University of California, Berkeley

Prof. Miquel Salmeron

Materials Science Division of the Lawrence Berkeley Laboratory
Materials Science and Engineering Department, University of California at Berkeley
Berkeley, California 94720, USA
mbsalmeron@lbl.gov • <http://stm.lbl.gov>

Berkeley, September 28, 2022

Dear editor:

We are happy to see the favorable response of the reviewers to our answer of their questions, which were very helpful to improve the paper.

Here is the response to the latest question by reviewer 1, regarding the equation on line 200 to quantify the ratio of Co^{2+} and Co^{3+} .

I hope that the answer is satisfactory and clarifies our calculation.

Response to the Reviewer 1:

Reviewer #1 (Remarks to the Author):

The authors have mostly addressed the comments properly.

Reply: We thank the reviewer for careful reading of the manuscript and for raising the constructive comments. Our point-by-point responses are listed below.

*However, questions regarding Figure 5 and the equation on line 312 remain. One can challenge the equation by applying it to defined systems where the $\text{Co}^{2+}/\text{Co}^{3+}$ concentration ratio is known, such as CoO with the spectra in Figure 5d. For these spectra, one gets $D(\text{Co}^{3+}) = \langle \text{small, negative} \rangle$ and $D(\text{Co}^{2+}) = \langle \text{comparably large} \rangle$. That is, $\text{RER} = D(\text{Co}^{3+})/D(\text{Co}^{2+}) = \langle \text{somewhat small, negative} \rangle$. With that one gets $N(\text{Co}^{3+})/N(\text{Co}^{2+}) = 2.99 * \text{RER}$. It is not zero as it should be. If one considers a hypothetical system with only Co^{3+} , then one may get a similar set of spectra as those in Figure 5d, with the Co^{2+} resonance spectrum replaced by the Co^{3+} resonance spectrum*

and the intensity ratio being inverted (i.e, the spectrum at the Co³⁺ resonance energy would be more intense than the spectrum at the Co²⁺ resonance energy). One would have D(Co³⁺)= and D(Co²⁺)=<comparably large> and RER=/<>comparably large>=<somewhat small>. That is, N(Co³⁺)/N(Co²⁺)=2.99<somewhat small>. However, it should be infinite. This seems to indicate that the equation is somewhat faulty.*

Reply: We thank the reviewer for critical comment about the ResPES.

As per validity of the formula, it uses a simple linear approximation to assess the Co²⁺ and Co³⁺ concentration ratio. Since the calibration point is at the stoichiometry of Co₃O₄, this is also a region where it will experimentally yield the best results. The validity interval of the formula is only between the two reference cases presented in the paper - CoO and Co₃O₄ – providing interpolation between these two stoichiometries, but no extrapolation.

$$\frac{N(\text{Co}^{3+})}{N(\text{Co}^{2+})} = y * \frac{D(\text{Co}^{3+})}{D(\text{Co}^{2+})} = y * RER$$

As for the case in the question for purely Co²⁺ present in the reference CoO sample. The D(Co³⁺) yields a small negative value of [0.1 (a.u.), measured from the plot in Fig.5(d)], caused by 2 factors. First is the fact that Co²⁺ 3d states exist throughout the entire valence band, including valence band maximum region. The D(Co³⁺) enhancement cannot therefore be completely disentangled from the Co²⁺ resonance, unless we use another, more sophisticated normalization. However, D(Co²⁺) yields dramatically larger enhancement of [~2.05 (a.u.), measured from the plot in Fig.5(d)], resulting in RER of ca. only ~4.8%, which is well below our experimental error. We comment on both of these limitations of the formula in the manuscript.

Proposed changes:

Page.8, Line 205-207: “Therefore, we can determine the concentration ratio, N(Co³⁺)/N(Co²⁺) of nonstoichiometric CoO_x film through the measurement of the RER if it contains both Co³⁺ and Co²⁺ sites”

Two details: ‘Res’ in Fig. 5e should probably be ‘RER’ and the colors of the labels in figs. 5d,e differ somewhat from the line colors (it is assumed that the authors wanted to have them identical).

Reply: We thank the reviewer for careful reading of the manuscript. The “Res” will be replaced by the “RER” in the Fig.5. For the line color (pure CoO) in **Fig.5d**, we want to make it different compared with **Fig.5e**. If the same line colors were used, the reader might be confused about the **Fig.5d** as to why the Co^{3+} -resonant spectrum (photon energy is 781.2eV) is measured in CoO since there is no Co^{3+} . Actually, for the Co^{2+} , this 781.2eV spectra is only off-resonant.

Proposed change:

Page 11, Fig.5 is replaced by the below image.

Response to the Reviewer 2:

Reviewer #2 (Remarks to the Author):

The authors have well addressed my comments. I recommend the publication of the manuscript on Nat. Commun.

Reply: We thank the reviewer for positive comments.

Response to the Reviewer 2:

Reviewer #3 (Remarks to the Author):

The authors properly addressed all questions. I recommend to accept the manuscript for publication in Nature Communications.

Reply: We appreciate the reviewer for positive comments.

Sincerely,

Miquel Salmeron